# Added body mass alters plantar shear stresses, postural control, and gait kinetics: Implications for obesity

Hwigeum Jeong[ID]*, A. Wayne Johnson, J. Brent Feland, Spencer R. Petersen, Jared M. Staten, Dustin A. Bruening[ID]°

Department of Exercise Science, Brigham Young University, Provo, Utah, United States of America

° These authors contributed equally to this work.
* hwiguem@gmail.com

**Data Availability Statement:** All data files are available from the OSF database (https://osf.io/hzfxj/).

## Abstract

### Context

Obesity is a growing global health concern. The increased body mass and altered mass distribution associated with obesity may be related to increases in plantar shear that putatively leads to physical functional deficits. Therefore, measurement of plantar shear may provide unique insights on the effects of body mass and body distribution on physical function or performance.

### Purpose

1) To investigate the effects of body mass and distribution on plantar shear. 2) To examine how altered plantar shear influences postural control and gait kinetics.

### Hypothesis

1) a weighted vest forward distributed (FV) would shift the center of pressure (CoP) location forward during standing compared with a weighted vest evenly distributed (EV), 2) FV would increase plantar shear spreading forces more than EV during standing, 3) FV would increase postural sway during standing while EV would not, and 4) FV would elicit greater compensatory changes during walking than EV.

### Methods

Twenty healthy young males participated in four different tests: 1) static test (for measuring plantar shear and CoP location without acceleration, 2) bilateral-foot standing postural control test, 3) single-foot standing postural test, and 4) walking test. All tests were executed in three different weight conditions: 1) unweighted (NV), 2) EV with 20% added body mass, and 3) FV, also with 20% added body mass. Plantar shear stresses were measured using a pressure/shear device, and several shear and postural control metrics were extracted. Repeated measures ANOVAs with Holms post hoc test were used to compare each metric among the three conditions ($\alpha = 0.05$).

**Funding:** The author(s) received no specific funding for this work.

**Competing interests:** The authors have declared that no competing interests exist.

## Results

FV and EV increased both AP and ML plantar shear forces compared to NV. FV shifted CoP forward in single-foot trials. FV and EV showed decreased CoP range and velocity and increased Time-to-Boundary (TTB) during postural control compared to NV. EV and FV showed increased breaking impulse and propulsive impulse compared to NV. In addition, EV showed even greater impulses than FV. While EV increased ML plantar shear spreading force, FV increased AP plantar shear spreading force during walking.

## Conclusion

Added body mass increases plantar shear spreading forces. Body mass distribution had greater effects during dynamic tasks. In addition, healthy young individuals seem to quickly adapt to external stimuli to control postural stability. However, as this is a first step study, follow-up studies are necessary to further support the clinical role of plantar shear in other populations such as elderly and individuals with obesity or diabetes.

## Introduction

Increased body mass inherently causes greater loading under the plantar surface of the foot during weight-bearing activity [1–3] Complications such as balance issues, falls and lower extremity injuries [4–6] in the obese population may be related to the altered biomechanics and plantar loading that arises from a chronically increased body mass. In addition, obesity typically positions the whole body center of mass (CoM) further anterior, altering loading distributions. This altered loading likely has a direct influence on obesity related complications such as postural instability [7] and increased fall risk [4], increased lower extremity muscle activity [8], altered gait mechanics [9], and reduced plantar cutaneous mechanoreceptor sensitivity [10]. Chronically increased loading may also cause foot deformities [11] and plantar tissue adaptations [11], such as increased skin thickness and hardness [12, 13], which may further reduce mechanoreceptor sensitivity and in turn further affect gait and balance.

Because body mass and body mass distribution by themselves can alter plantar loads, isolating and studying their acute effects on plantar loading may be helpful in understanding the mechanisms behind biomechanical complications in the obese population. For instance, non-obese individuals with added loads show increased plantar pressure [14] and reduced plantar cutaneous sensitivity [15]. The latter is likely due to Weber's law, where increased pressure stimulus results in a higher mechanoreceptor detection threshold [16]. However, increased body mass alone does not fully explain the increased postural sway and altered gait mechanics in individuals with obesity [17–20]. For example, non-obese individuals wearing an evenly distributed load have shown similar [21] or even decreased postural sway [20] compared to wearing an unevenly distributed load, which increases sway [20, 21], particularly in somatosensory compromised environments (e.g. eyes closed) [22]. Although previous studies have investigated the effects of added body mass and altered mass distribution on postural control and gait mechanics, these have focused almost exclusively on load carriage applications, such as military personnel [20] or students who carry heavy backpacks [21]. To our knowledge, no studies have used added body mass to simulate the forward mass distribution that occurs with obesity.

Our feet are impacted not only by vertical pressure, but also by horizontal shear stresses. While the net shear components of the ground reaction force are relatively much lower than

the vertical component, they may have a large influence on plantar tissue breakdown and mechanoreceptor sensitivity. Plantar tissue spreading is likely increased with added body mass, as evidenced by a greater foot contact area in individuals with obesity [2]—this spreading may be more apparent when analyzing shear stresses than pressure. In addition, the foot's sole contains numerous fast adapting mechanoreceptors [23], which are sensitive to small changes in stimuli. Changes in shear stresses can be similar to changes in vertical pressure, thus having a relatively higher change compared to baseline values [24]. In addition, net shear forces are directly proportional to CoM-CoP differences and therefore carry phasic sway-relevant information needed for balance [24, 25]. While the importance of shear stresses on mechanoreceptor function and postural control was recognized two decades ago [25], shear distribution measurements have been hampered by technological limitations. Recent advances in shear sensing technology have enabled accurate measurement of shear stress [26] and may provide us with a first look at how added body mass affects plantar shear stresses.

The purpose of this study was to investigate the effects of added body mass and distribution on plantar shear stresses in non-obese individuals, both in standing and walking. We also sought to preliminarily assess their effects on postural stability. We hypothesized that 1) a front-loaded weighted vest (FV) would shift the CoP location forward during standing compared to an evenly distributed weighted vest (EV), 2) FV would increase plantar shear spreading forces more than EV during standing, 3) FV would increase postural sway during standing while EV would not, and 4) FV would elicit greater compensatory changes during walking than EV, resulting in altered plantar impulses and plantar shear spreading forces.

## Methods

### Participants

A sample of 20 young healthy male participants were recruited (age = 23 ± 3.10 years; height = 180 ± 0.05 cm; mass = 75 ± 8.02 kg; BMI = 23 ± 1.85 kg/m$^2$). In consideration of the distinct body fat accumulation tendencies [27] as well as movement differences [28] between sexes, we recruited only male participants. Young adults were chosen to minimize the influence of altered muscle strength [29] or spatiotemporal gait factors in older age groups [30]. Inclusion criteria consisted of normal body weight—BMI: 20–25% defined by World Health Organization (WHO) [31]. Participants were screened using the Lower Extremity Functional Scale (LEFS) [32] and excluded if they had issues with the lower limb musculoskeletal system affecting mobility or balance such as symptomatic osteoarthritis, any known heart disease, any neurological conditions such as diabetes, or lower and upper extremity injuries in the past 6 months. Participants also had to be able to stand and walk unassisted with an added load. This study was conducted in accordance with the declaration of Helsinki. All participants were volunteers and signed an informed consent form approved by the Brigham Young University Institutional Review Board (Protocol # F19248).

### Protocol

Participants performed four tests in the following order: 1 –static stance, 2 –bilateral-foot postural control, 3– single-foot postural control, and 4 –walking. Three different weight conditions were used in each test: Unweighted (NV), evenly distributed weighted vest (EV), and front-loaded weighted vest (FV) (Fig 1). The order of the weight conditions was randomized within each test. For both weighted conditions, 20% of the participant's body mass was added, in the form of individuals 0.45 kg sand bags. The 20% added weight increased mean BMI to 27.80 (± 2.22 kg/m$^2$). According to WHO's obesity definition [31], this would classify the participants with the added weight in the range of overweight. For EV, weights were added equally

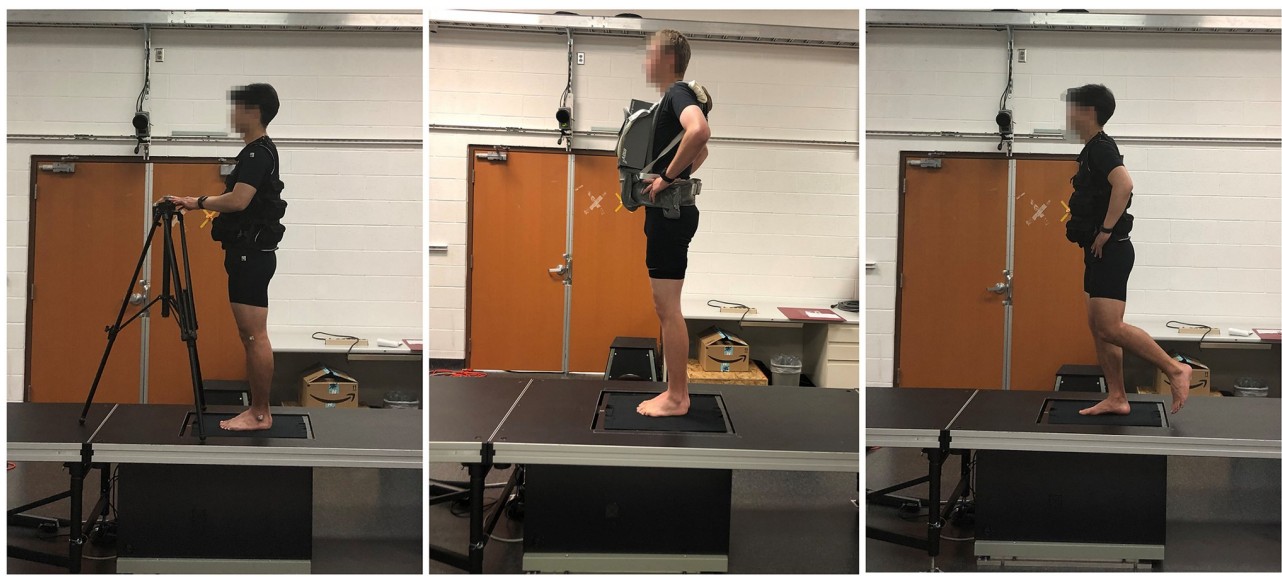

**Fig 1. Raised platform/walkway, weight conditions, and standing tests.** Left: Static trial with evenly distributed (EV) weight. Middle: Bilateral-foot standing trial with front weight (FV). Right: Single-foot standing trial with EV.

to the front and back of an exercise vest (Valeo VA4471, Houston, TX USA), while for FV, the same amount of weight was added to the front of the body using a baby carrier (Sunveno sv22094, New Delhi India) (Fig 1). The baby carrier was used in place of the exercise vest because it better accommodated an asymmetrical load and was more comfortable around the shoulders and neck when fully front loaded. Participants were given several minutes to acclimate to the vest conditions prior to testing. In addition, to minimize muscle fatigue, participants were provided with a 1-minute seated rest period between each trial and a 3-minute seated rest period between each test.

A pressure and shear measurement device (FootSTEPS, Innovative Scientific Solutions, Inc., Dayton OH, USA) was used to collect all plantar force data, sampling at 25 Hz. for standing trials and 50 Hz. for walking. Details regarding the device hardware and measurement validity have been published previously [26]. Briefly, the device consists of a glass plate with an embedded stress sensitive polymer film (0.42 by 0.28 m area). A camera underneath the plate captures film displacements which are converted to vertical pressure and mediolateral and anteroposterior shear stress distributions using a finite element reconstruction model. A force plate (AMTI, Watertown MA, USA) mounted underneath the device is used for calibration. Adjustable height staging panels (StageRight Z-HD, Clare, MI, USA) were used to make a platform and walkway that was flush with the sensing surface (Fig 1). A hole was cut in the center panel for the sensor and a small (< 1 cm) gap was maintained around the device perimeter.

The static stance test was used to evaluate the influence of added mass on plantar forces without the confounding influences of the small accelerations that can accompany balance tests. Thus, this test consisted of an assisted standing posture. The participant stood barefoot in front of the pressure/shear sensor. Once given a starting sign from a researcher, the participant took a step onto the sensor. The participant was instructed to initially step his dominant foot on the sensor and then to sequentially settle the opposite foot next to the dominant foot. Assistance was provided in the form of a fixed bar placed in front of the sensor. Once putting feet together on the sensor, the participant was instructed to lightly touch the bar only with his fingertips, but not to grasp, push, or pull (Fig 1). To standardize posture as well as minimize

reliance on the bar, the participant was instructed to watch a screen in front of the participant that showed the participant in profile superimposed on a grid. The grid was provided by a motion capture system (Qualisys, Inc., Gothenburg, Sweden) linked to the screen. Four reflective markers were used to enable the participant to maintain an upright standing position. These markers were attached to the lateral shoulder, greater trochanter, femoral epicondyle, and malleolus on the left side, forming a line. The participant was asked to maintain alignment of these four markers on a line of the grid in order to maintain consistent upright position during the test. Three trials of 25 seconds each were recorded. To avoid extraneous movements at the beginning and end of a trial, we removed the first 7 and last 5 seconds, thus, the actual recording period was 13 seconds.

For both postural control tests (Fig 1), the participant took one step onto the sensor once a researcher gave a starting sign, placed hands on iliac crests, and stood as still as possible. The bar and grid were not used, but the participant was instructed to visually focus on a marked area at approximately eye level. For the 2-foot tests, the participant was instructed to place both feet so that they were touching, in as narrow a stance as possible. In the single-foot standing, a participant stood on his dominant foot, bending his opposite knee in order to avoid the foot touching the ground. Three successful trials of 25 seconds each were recorded for each test. Trials were discarded if hands were taken off the hip or the toes of the opposite leg touched the ground (in the single-foot trials) during the recording. Again, the first 7 and last 5 seconds were also removed prior to processing.

In the walking test, preferred walking speed was first determined in order to minimize confounding factors, considering that walking speed by itself affects gait mechanics [33]. While a participant walked barefoot back and forth on the walkway (5.5 m long and 1.0 m wide), average walking speed was measured using laser timers placed near each end of the middle third of the walkway. The same laser timers were used to monitor walking speed during the actual test. The participant was instructed to walk as normal as possible and to keep looking straight ahead. The starting position was adjusted by a researcher to ensure full contact of the dominant foot on the sensing area. Moreover, the participant received verbal feedback after a walking trial in order to maintain the walking speed within ± 10% of his average walking speed. Three successful trials (i.e. clean foot contacts at average walking speed) were collected.

## Data analysis

Data from the pressure/shear device was first processed in manufacturer supplied software, as described previously [26]. Raw shear stress and reconstructed pressure data were then imported into custom LabView software (National Instruments, Austin TX, USA) for data analysis.

For static stance, mean AP and ML plantar spreading shear forces and AP CoP location were extracted. To determine plantar spreading forces, all directional shear stresses were summed separately and multiplied by the pixel areas to create directional shear GRFs. Specifically, anterior directed shear stresses were separated from posterior directed shear stresses; likewise, medial and lateral shear stresses were separated from each other. AP and ML spreading was then determined as the amount of each directional force that opposed the main directional force. For example, if anterior force exceeded posterior force, the absolute value of the posterior force was used to represent plantar spreading. CoP was calculated as the weighted average of all pressure pixel locations ($x_i$), weighted by vertical force ($F_{vi}$) (Eq (1)). This was expressed in the AP direction as a percentage of foot length, relative to the heel. To do this, a composite plantar pressure image was constructed (max value of each pixel across time) and the maximum anterior and posterior boundaries of the footprint were identified using

pressure thresholds.

$$CoP_{AP} = \frac{\sum (F_{vi} * x_i)}{\sum F_{vi}} \tag{1}$$

For the postural control tests, the following dependent variables were extracted: AP CoP location, mean CoP speed, AP and ML CoP range, and AP and ML TTB. These measures were chosen to represent a broad range of previously published metrics from the postural control literature. CoP location was calculated in the same way as it was in static stance. Mean CoP speed was obtained by summing instantaneous CoP displacements (i.e. path length) and dividing by total trial duration (13 seconds). CoP range was represented as the difference between maximum and minimum CoP values in both AP and ML directions. TTB was calculated as previously described for both AP and ML directions [34]. Briefly, rectangular AP and ML boundaries were identified from a composite pressure image. The distance between the boundary and the instantaneous CoP location were divided by instantaneous velocity (central difference) in that direction. For example, if CoP was moving toward the metatarsal heads or anterior boundary, the distance between the anterior boundary and the CoP was divided by the instantaneous velocities which corresponded to the anterior direction. AP and ML TTB(s) were obtained separately. From the TTB time series, we calculated the mean of all local signal minima.

For the walking test, the following dependent variables were extracted: braking (posterior) impulse, propulsive (anterior) impulse, and AP and ML plantar shear spreading at midstance. Braking and propulsive impulses were obtained from the net AP forces using trapezoidal integration. For braking impulse, the area from heel strike to midstance was obtained, while for propulsive impulse, the area from midstance to toe-off was obtained. AP and ML plantar spreading forces were calculated in the same way as was done for the static test, but extracted at midstance rather than taking the mean across stance. All metrics were averaged across trials.

### Statistical analysis

All metrics were compared across weight conditions using repeated measures analysis of variance (ANOVA). For statistically significant main effects ($p < 0.05$), a post hoc test, Holm method, was used for pairwise comparisons. In addition, eta squared effect size was calculated ($\eta^2$). Based on Cohen's guideline, eta effect size was defined as small ($\eta^2 = 0.01$), medium ($\eta^2 = 0.06$), and large ($\eta^2 = 0.14$) effects [35]. Mauchly's test was used to check sphericity violation. If sphericity was violated, Greenhouse-Geisser test or Huynh-Feldt test was used to correct the violation depending on 0.75 of the epsilon values. If the epsilon is bigger than 0.75, Greenhouse-Geisser test was used to correct sphericity violation. Otherwise, Huynh-Feldt was selected for correction of sphericity violation. Furthermore, the Benjamini–Hochberg procedure with a false-discovery rate of 0.10 was used on the ANOVA main effects to account for the multiple tests performed.

### Results

Of the 20 total metrics analyzed (3 static, 12 postural control, 5 walking), 17 showed significant main effects due to the weight conditions (Tables 1–3). One of these (AP CoP range bilateral-foot) had a p-value greater than 0.05 (0.058), but was identified as significant by the Benjamini Hochberg procedure.

**Table 1. Descriptive information from static test (Mean ± SD).**

|  | FV | NV | EV | *P*-value ($\eta^2$) |
|---|---|---|---|---|
| **AP Plantar Shear Force (N)** | 27.8 ± 6.2* | 23.0 ± 6.4 | 29.7 ± 5.9* | < 0.001 (0.45) |
| **ML Plantar Shear Force (N)** | 41.9 ± 8.4* | 31.2 ± 10.0 | 39.1 ± 7.6* | < 0.001 (0.47) |
| **AP CoP Location (%)** | 47.2 ± 6.0 | 45.4 ± 5.7 | 46.88 ± 5.2 | 0.176 (0.09) |

*P*-value is ANOVA main effect.

* Significant compared to NV.

**Table 2. Descriptive information of postural control test¶ (Mean ± SD).**

|  | Bf_FV | Bf_NV | Bf_EV | *p*-value of Bf ($\eta^2$) | Sf_FV | Sf_NV | Sf_EV | *p*-value of Sf ($\eta^2$) |
|---|---|---|---|---|---|---|---|---|
| **AP CoP location (%)** | 44.7 ± 6.1 | 45.5 ± 6.1 | 44.4 ± 6.6 | 0.360 (0.05) | 51.6 ± 3.1*+ | 49.9 ± 2.9 | 49.9 ± 3.4 | 0.011 (0.21) |
| **CoP Speed (cm/sec)** | 5.4 ± 1.2 | 7.3 ± 1.6+¶ | 5.8 ± 1.3¶ | < 0.001 (0.78) | 7.0 ± 1.7 | 7.8 ± 2.6+¶ | 6.8 ± 2.1 | 0.012 (0.21) |
| **AP CoP Range (cm)** | 2.5 ± 0.5 | 2.8 ± 0.5 | 2.8 ± 0.6 | 0.058§ (0.14) | 3.6 ± 0.6 | 3.8 ± 0.6 | 3.7 ± 0.7 | 0.392 (0.05) |
| **ML CoP Range (cm)** | 2.9 ± 0.7 | 3.3 ± 0.7*¶ | 3.1 ± 0.5 | 0.002 (0.24) | 3.7 ± 0.6 | 4.3 ± 0.9*¶ | 3.8 ± 0.9 | 0.005 (0.28) |
| **AP TTB (sec)** | 4.6 ± 1.4* | 3.3 ± 0.9 | 4.4 ± 1.5* | < 0.001 (0.60) | 3.3 ± 0.9 | 3.0 ± 0.7 | 3.5 ± 0.7* | 0.015 (0.20) |
| **ML TTB (sec)** | 2.4 ± 0.4* | 1.8 ± 0.3 | 2.3 ± 0.3* | < 0.001 (0.76) | 1.1 ± 0.3 | 0.9 ± 0.3 | 1.1 ± 0.4* | 0.029 (0.17) |

Bf is bilateral-foot. Sf is single-foot. *p*-value is main effect.

* Significant compared to NV.

¶ Significant compared to FV.

+ Significant compared to EV.

§ Main effects from Benjamini-Hochberg procedure despite *p* > .05.

## Static test

ANOVA showed main effects on plantar spreading forces in both AP ($p < .001$) and ML ($p < .001$) directions. The pairwise comparisons revealed that both AP and ML plantar shear forces were greater in FV and EV than in NV while differences were not seen between FV and EV (Fig 2). There were no differences in CoP location between the weight conditions ($p = 0.176$) (Fig 2). Table 1 provides the outcomes of the static test in summary.

## Postural control tests

As with CoP location in the static test, CoP location in the bilateral-foot trials showed no differences ($p = .36$) (Fig 3). However, there were main effects on CoP location in the single-foot

**Table 3. Descriptive information of walking test (Mean ± SD).**

|  | FV | NV | EV | *P*-value ($\eta^2$) |
|---|---|---|---|---|
| **Braking Impulse (N•sec)** | 9.5 ± 2.4* | 8.1 ± 2.2 | 10.1 ± 2.4*¶ | < 0.001 (0.61) |
| **Propulsive Impulse (N•sec)** | 10.7 ± 2.1* | 9.0 ± 1.5 | 11.6 ± 2.2*¶ | < 0.001 (0.69) |
| **AP Spread at Midstance (N)** | 31.3 ± 5.6*+ | 25.5 ± 4.3 | 29.0 ± 5.6* | < 0.001 (0.48) |
| **ML Spread at Midstance (N)** | 21.4 ± 5.2* | 18.1 ± 4.0 | 23.7 ± 4.8*¶ | < 0.001 (0.55) |

*p*-value is main effect.

* Significant compared to NV.

¶ Significant compared to FV.

+ Significant compared to EV.

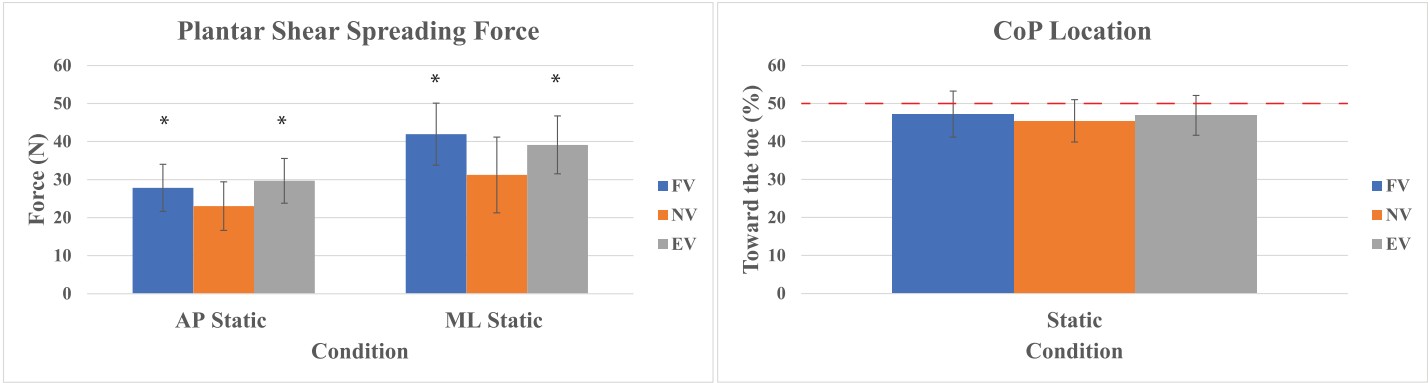

**Fig 2. Static AP & ML plantar shear spreading and static CoP location % with standard deviations.** The horizontal red line is the central position (50% of a foot length). * Significant compared to NV.

trials ($p = .011$), with the pairwise comparisons revealing that the CoP location was more anterior in FV as compared to NV and EV (Fig 3). The statistical analysis showed main effects on CoP speed in bilateral-foot trials ($p < .001$) and single-foot trials ($p = .012$) with FV and EV decreased compared to NV, and FV further reduced compared to EV just in the bilateral-foot trials (Fig 3).

The Benjamini-Hochberg procedure found main effects on AP CoP range in bilateral-foot trials ($p = .058$), but no pairwise comparisons were significant (Fig 4). No main effects were noted in AP CoP range in single-foot trials ($p = .392$). Meanwhile, ANOVA showed main effects on ML CoP range in bilateral-foot trials ($p = .002$) and in single-foot trials ($p = .005$). The post hoc tests revealed that FV and EV decreased ML CoP range in comparison to NV (Fig 4).

For AP TTB, ANOVA found main effects in bilateral-foot trials ($p < .001$) and in single-foot trials ($p = .015$). The post hoc tests observed that FV and EV increased AP TTB in bilateral-foot trials as compared to NV while only EV showed an increase in AP TTB in single-foot trials compared to NV (Fig 5). Likewise, the statistical analysis showed main effects on ML TTB in bilateral-foot trials ($p < .001$) and in single-foot trials ($p = .029$). The post hoc tests observed that, similar to the outcomes of AP TTB, FV and EV increased ML TTB in bilateral-

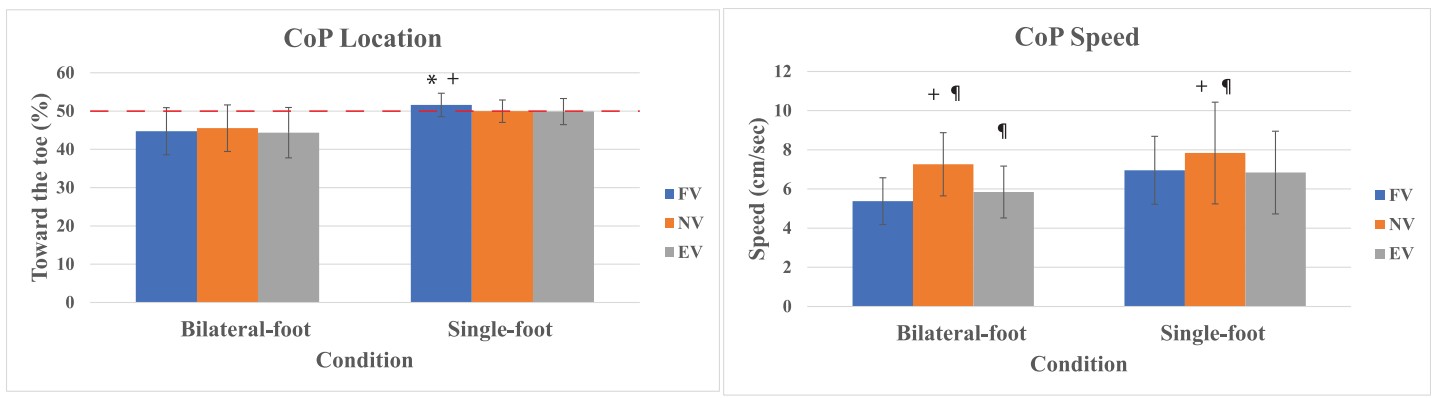

**Fig 3. Standing CoP location% in bilateral-foot trials and single-foot trials and CoP speed with standard deviation.** The horizontal red line is the central position (50% of a foot length). * Significant compared to NV. ⸻ Significant compared to FV. + Significant compared to EV.

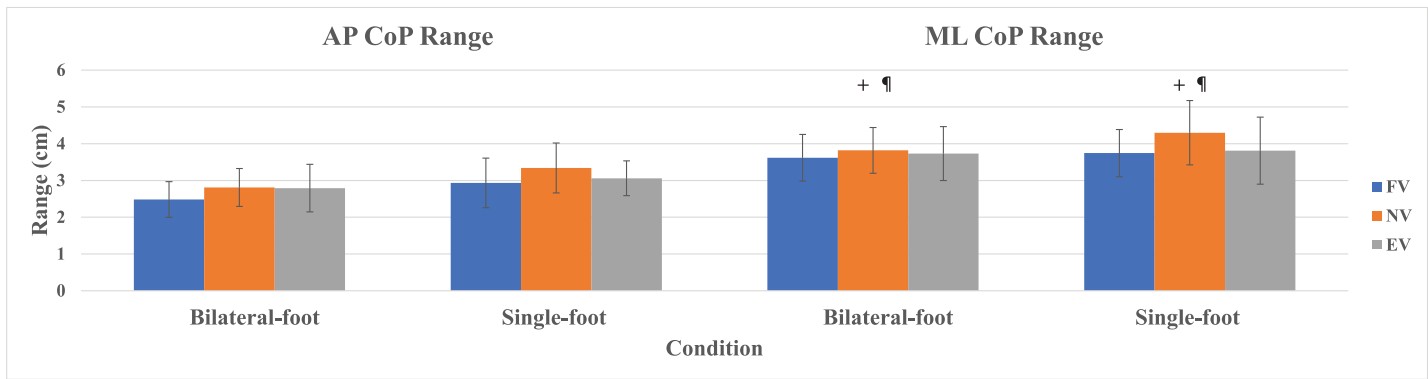

**Fig 4. Standing AP & ML foot CoP ranges with standard deviation.** ¶ Significant compared to FV. + Significant compared to EV.

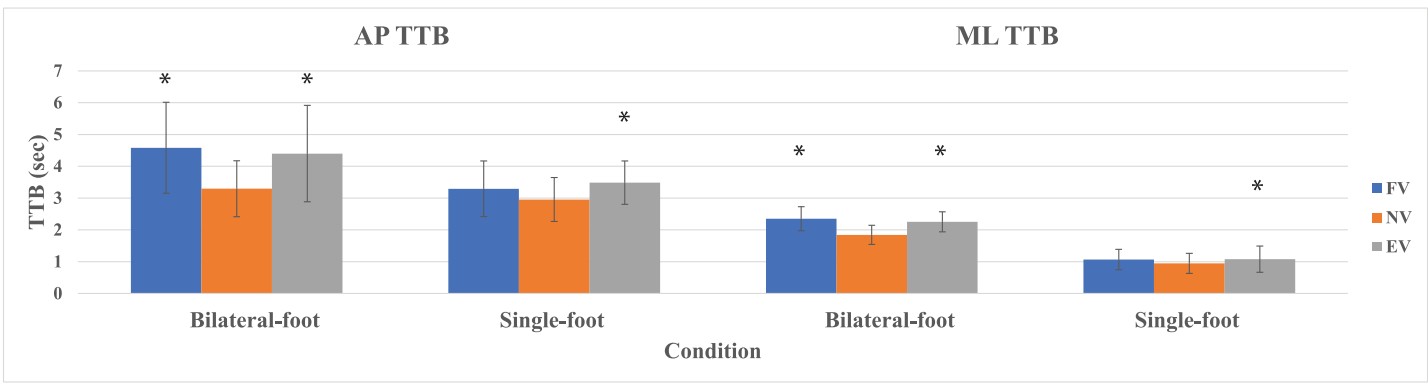

**Fig 5. Standing AP & ML TTB with standard deviation.** * Significant compared to NV.

foot trials as compared to NV, while only EV showed an increase in AP TTB in single-foot trials compared to NV (Fig 5). Table 2 provides the outcomes of the standing test in summary.

## Walking test

The average walking speed was 1.22 m/s (± 0.28). Main effects were found for both braking and propulsive impulses ($p < .001$, $p < .001$, respectively) (Fig 6). According to the pairwise

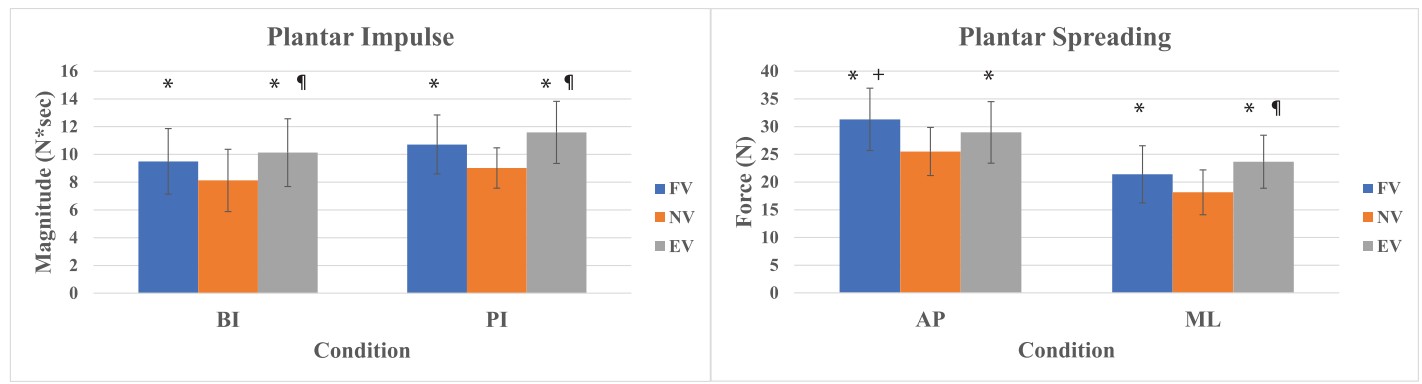

**Fig 6. Walking peak braking force, plantar braking impulse (BI) & propulsive impulse (PI), and AP & ML plantar spreading with standard deviation.** * Significant compared to NV. ¶ Significant compared to FV. + Significant compared to EV.

comparisons, FV and EV increased braking impulse and propulsive impulse compared to NV. Meanwhile, EV showed further increase in braking impulse and propulsive impulse as compared to FV (Fig 6). For plantar shear spreading forces, there were main effects in both AP ($p < .001$) and ML ($p < .001$), respectively. Based on the pairwise comparisons, FV and EV increased AP plantar shear forces and ML plantar shear forces as compared to NV. Interestingly, while AP plantar shear spreading forces were even greater in FV than in EV, ML plantar shear forces were conversely greater in EV than in FV (Fig 6). Table 3 provides the outcomes of the walking test in summary.

## Discussion

### Static test

The purpose of the static test was to isolate the effects of added body mass on plantar loading without balance related influences. As expected, added body mass resulted in increased plantar spreading in both AP and ML directions. However, there were no differences in spreading between FV and EV conditions, nor did we see a change in CoP location in the static tests. It is possible that the weight distribution in the FV condition did not change the CoM far enough anterior to see larger effects on these variables. Alternately, the controlled posture and hand assistance may have resulted in elevated activity of the neuromuscular system in order to maintain the upright position.

While elevated plantar spreading was expected, this study represents the first measurement of plantar shear forces in this context. Plantar spreading forces were slightly increased above the 20% increase in net vertical force, with ML spreading (34% FV and 25% EV) slightly greater than AP spreading (21% FV and 29% EV), but both suggestive of plantar skin shearing as well as deforming effects on the transverse and medial longitudinal foot arches. A few previous studies have indirectly investigated the effects of mass on medial longitudinal arch deformation [36–38]. Comparing seated and standing arch height, Butler at al. reported a 3 mm change in arch height [36] while Xiong et al. reported a 6% change in foot length [37]. Wright et al. added 10 kg to the knee of seated participants, measuring a 1 mm drop in arch height that became greater as the tibia moved anterior [38]. However, these studies examined normal weight loading, rather than changes due to loading beyond normal body weight. In contrast, Kern et al. used a weighted vest during walking, finding no difference in arch height with added mass [39]. Yet, obese individuals have lower arches [40]—this plantar spreading could represent a mechanism for long term arch collapse.

### Postural control tests

Similar to the CoP location in the static test, the CoP location in the bilateral-foot trials did not show any differences among conditions. However, in the more challenging single-foot trials, FV elicited a 3% forward shift in CoP. In single-foot standing, the participants were instructed to slightly bend the knee of the support leg in order to avoid confounding effects, such as joint locking. The flexed knee may have prevented any compensatory trunk extension in FV.

The increased plantar pressure and shear spreading that was quantified in the static test may negatively influence standing postural control. Tactile perception thresholds appear to increase with elevated pressure, in both obese [7] and healthy individuals (after standing and walking with added mass) [15], suggesting reduced mechanoreceptor sensitivity with increased sensory feedback (i.e. Weber's law) [16]. While no studies have investigated the relationship between plantar shear stresses and plantar sensitivity, mechanoreceptors are expected to receive contributions from both pressure and shear. This reduced plantar cutaneous

sensitivity has been associated with increased postural sway [7], thus driving our decision to simultaneously investigate postural sway measures.

In contrast to our hypothesis, FV showed no differences in postural control metrics compared to EV. Furthermore, both EV and FV consistently showed lower CoP range, lower CoP speed, and greater TTB(s) than NV. While this was not completely unexpected in EV, previous studies utilizing backpacks have shown increased postural sway [21]. It is likely that in our study the FV condition did not displace the CoM anteriorly as much as a heavy backpack can displace CoM posteriorly. Yet, this condition is much more similar to CoM changes seen in obesity. The changes in postural control metrics seen in both conditions are traditionally thought to represent an increase in postural stability. However, our results and others suggest that this perspective should be interpreted cautiously, as reduced movement variability does not always indicate healthy or good postural stability, particularly in young, asymptomatic individuals [41]. For example, one previous study observed that young participants demonstrated increased variability when placed in a neutral posture as compared to more challenging AP leaning postures while older participants exhibited the reverse [34]. Healthy young individuals have a sensitive neuromuscular system that can easily detect changes in stimuli and alter neuromuscular function to compensate [42]. Both weighted conditions may have stimulated the participants to focus more intently on achieving postural equilibrium through sensory reweighting and/or increased muscle activation [43]. Young healthy subjects may be able to more easily utilize other body systems (e.g., proprioception, visual, and vestibular systems) in order to compensate for degraded plantar mechanoreceptor sensitivity. For instance, there is evidence of sensory reweighting in a disruptive injury such as chronic ankle instability [43]. In addition, quick reflex-induced postural reactions may have compensated for the greater ankle joint torques induced by the extra load [44]. In other words, the seemingly increased postural stability in FV and EV may result from the ability to flexibly adapt to changing conditions, rather than representing real improved postural stability [41]. In the current study, much greater increases in mass and forward positioning are likely needed to truly challenge postural control in this population. However, pilot testing suggested that this level of added mass (i.e. obesity instead of overweight) would be difficult to achieve without discomfort inducing additional postural compensations (e.g. too much load on the vest straps). To truly simulate the effects of obesity on postural control, follow-up studies would require novel methodology to add body mass and distribution to reach obesity without any discomfort.

## Walking test

Walking was included in this study to evaluate the manner in which added body mass and distribution affects plantar loading during a common dynamic movement, and how this might ultimately influence postural control. As expected, we found strong influences in all loading metrics from both conditions compared to NV. Previous studies on both load carriage and obesity have shown that when using absolute force, most plantar loading metrics increase; however, when normalized to body mass these differences disappear [45, 46]. We did not normalize to total body mass as our primary purpose was to compare between loaded conditions. In all four chosen metrics, we saw differences between FV and EV.

Both braking and propulsive impulses were lower in FV than EV. These differences appear to be consistent with greater postural instability in FV than EV during the dynamic tasks. While postural instability was not apparent in the standing postural control tests, dynamic movements like walking should be much harder to adapt to. Previous load carriage studies show that healthy young individuals walking with a heavy backpack decrease stride length and increase double support time by decreasing hip and knee sagittal plane angular displacement

[47]. Although we did not analyze kinematics, we suspect that FV elicits similar compensations with a purpose of increasing dynamic postural stability. For example, shortening stride length allows for a more stable position at initial contact with a more forward CoP and shorter single support time, theoretically resulting in decreased braking and propulsive impulses. In contrast, the more stable EV condition should elicit these compensations to a lesser degree. In addition, a heavy backpack tends to alter upper extremity posture, resulting in greater trunk flexion with accompanying elevated back muscle activity [48]. The differences in both braking and propulsive impulses may be due to altered upper extremity posture, potentially with FV hyperextending their trunk during walking in order to compensate for a more forward CoM. For future studies, adding kinematic variables will be helpful to better understand the relationship between body distribution and impulses.

Unlike walking impulses, plantar shear spreading forces showed different interactive effects between body mass and body location. As expected, FV and EV both showed greater plantar shear forces than NV. Interestingly, however, EV increased ML plantar shear spreading forces more than FV, while conversely, FV increased AP plantar shear spreading forces more than EV. The increased AP spreading forces in FV occur during midstance, and may be due to a more forward positioned CoP and greater midfoot torque. While we cannot confirm this mechanism in this study, this increased AP spreading does indicate additional stress on the medial longitudinal arch of the foot and may be a contributing mechanism to altered foot morphology over time. This should be further investigated. It is not clear to us why ML spreading stresses were conversely higher in EV; we can only speculate potential kinematic changes such as stride width. This dichotomy requires further investigation. While we did not measure foot kinematics, Kern et al. did not show any differences in midfoot kinematics with added body mass (similar to our EV condition) [39]. Our results suggest that plantar shear forces may be altered without noticeable midfoot kinematics and eventually lead to damage of soft tissues over time. In addition, these stresses appear to be dependent on body mass distribution. However, to clarify the effects of plantar shear forces on foot morphology, follow-up studies are necessary.

In both FV and EV, plantar force metrics are elevated above NV, with several clinical implications. In addition to concerns over postural control and falling, greater loading in general may increase the incidence of other pathologies such as osteoarthritis [49]. As mentioned above, the increased plantar shear spreading forces in both FV and EV likely also alters plantar mechanoreceptor sensitivity. This may be a critical concern in neurological pathologies such as diabetes, where neuropathy may potentially be influenced by altered shear spreading [50]. Body mass distribution may be also an important factor in these pathologies. For example, one study reported that after walking with a backpack (30% of body weight) for just 10 minutes, healthy young participants showed reduced somatosensory function concurrently with increased AP postural sway [51]. However, the participants did not alter somatosensory function after walking with a double-pack that was the same weight as the backpack. To authenticate the relationship between plantar shear and plantar mechanoreceptor sensitivity, more information is needed from future studies.

Finally, the increased plantar shear spreading could also represent altered energetics. Increased plantar shear stresses are likely associated with increased energy dissipation in the form of heat, which could reduce walking efficiency even beyond that due solely to the added energy need to raise and propel body mass forward [52]. This may make the proportionally lower efficiency in obesity even more impressive [53]. Of course, the full contribution of shear stresses to walking energetics requires additional research.

## Conclusions

Elevated body mass increased plantar shear spreading forces in both standing and walking. This did not result in decreased standing postural control, likely due to the use of healthy young participants, who can flexibly adapt to acute external stimuli in a simple standing task. However, differences between loading conditions in the walking tasks suggest that body mass distribution does influence plantar loading and likely dynamic postural control. The effects of body weight and distribution on postural control may depend on the task difficulty (static vs. dynamic task). This study represents the first measurement of plantar shear stresses in the context of altered body mass, and helps identify future study directions. Specifically, follow-up studies are necessary in other populations, such as obese individuals who chronically carry excessive mass in the abdomen. Additionally, since we exclusively tested young male participants, our results may not fully generalized to females or elderly individuals, and factors related to these populations should also be considered. Furthermore, integrating kinetic measurements with kinematic and neuromuscular variables will provide further insights on how plantar shear influences dynamic tasks like gait.

## Acknowledgments

We would like to thank Jordan Grover for his help with data collection.

## Author Contributions

**Conceptualization:** Hwigeum Jeong, A. Wayne Johnson, J. Brent Feland, Dustin A. Bruening.

**Data curation:** Hwigeum Jeong, Jared M. Staten.

**Formal analysis:** Hwigeum Jeong, A. Wayne Johnson, J. Brent Feland, Spencer R. Petersen, Dustin A. Bruening.

**Investigation:** Hwigeum Jeong, Jared M. Staten, Dustin A. Bruening.

**Methodology:** Hwigeum Jeong, A. Wayne Johnson, J. Brent Feland, Dustin A. Bruening.

**Software:** Spencer R. Petersen.

**Supervision:** Dustin A. Bruening.

**Writing – original draft:** Hwigeum Jeong, A. Wayne Johnson, J. Brent Feland, Dustin A. Bruening.

**Writing – review & editing:** Hwigeum Jeong, A. Wayne Johnson, J. Brent Feland, Spencer R. Petersen, Jared M. Staten, Dustin A. Bruening.

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
