## [Decision Letter · Decision Letter 0]

1 Dec 2020

PONE-D-20-30951

Added body mass alters plantar shear stresses, postural control, and gait Kinetics: Implications for obesity.

PLOS ONE

Dear Dr. Jeong,

Thank you for submitting your manuscript to PLOS ONE. After careful consideration, we feel that it has merit but does not fully meet PLOS ONE’s publication criteria as it currently stands. Therefore, we invite you to submit a revised version of the manuscript that addresses the points raised during the review process.

We look forward to receiving your revised manuscript.

Kind regards,

Bijan Najafi

Academic Editor

PLOS ONE

Journal Requirements:

Reviewers' comments:

Reviewer's Responses to Questions

**Comments to the Author**

1. Is the manuscript technically sound, and do the data support the conclusions?

Reviewer #1: Yes

Reviewer #2: Yes

2. Has the statistical analysis been performed appropriately and rigorously? 

Reviewer #1: Yes

Reviewer #2: Yes

3. Have the authors made all data underlying the findings in their manuscript fully available?

Reviewer #1: Yes

Reviewer #2: Yes

4. Is the manuscript presented in an intelligible fashion and written in standard English?

Reviewer #1: Yes

Reviewer #2: Yes

5. Review Comments to the Author

Reviewer #1: The authors present an interesting study that investigated the effect of increased body mass and altered mass distribution associated with obesity on plantar shear that may putatively lead to physical functional deficits. While the results are potentially informative and may provide a meaningful contribution to this rapidly developing field, there were a few aspects of the paper that were unclear as currently written.

1. Authors justified of excluding female based on distinct body fat accumulation tendency between male and female. However, in the current study objective of the authors is to compare effect of weight distribution conditions (i.e., Forward distributed vs Evenly distributed vs Normal weight) and not within subject differences. Therefore, in my opinion given justification to exclude female is weak and raises issues related to generalizability of the results.

2. Authors need to double check for the typo. For instance, in the abstract, there were typo such as (..).

3. Once the abbreviation of a term is provided than authors do not need to provide it again. For instance in the abstract authors provided abbreviation for evenly distributed (EV), or frontloaded vest (FV) twice.

4. In the methodology, if the experiments were conducted according to declaration of Helsinki then authors should mention it in the methodology.

5. In the figure, significant differences are not represented. The figures should be self-explanatory, and readers should not be needed to refer text or able to identify group significant differences. Providing significant differences through arrow would make it convenient for the readers to understand. Additionally, in figure 1, authors should have represented the AP_Static or ML_Static as AP Static or ML Static.

6. In discussion, references are required for the statement, “A few previous studies have indirectly investigated the effects of mass on medial longitudinal arch deformation.”

7. The following sentence in the discussion was confusing and need to be rephrashed, “This reduced plantar cutaneous sensitivity has been associated with increased postural sway [7], thus our decision to simultaneously investigate postural sway measures.”

8. It was not clear if the authors want to indicate association or causation in the following statement in discussion, “Previous load carriage studies show that healthy young individuals walking with a heavy backpack decrease stride length and increase double support time through decreased hip and knee sagittal plane angular displacement [44].”

Reviewer #2: Abstract

-“However, the interactive effects between body mass and distribution may disrupt physical function and/or performance in other populations—such as elderly and individuals with obesity or diabetes.” I’m not sure this speculation is needed in the abstract.

Introduction

-“Changes in shear stresses are relatively higher compared to baseline values than changes in vertical pressure, and thus shear detecting mechanoreceptors may have a greater influence on relaying postural sway information needed for balance.” This sentence is not clear. Under what conditions or in what situations are changes in shear stresses expected to be relatively higher? Also are there any references that can be cited to support the premise the authors are trying to convey?

Methods

-Participants: The first sentence reads as though a sample of participants with actual ages ranging from 18 to 40 years old were recruited. However, the 3rd sentence seems to suggest the previously named age range was an inclusion criterion as opposed to a description of the actual participants enrolled. Please clarify what 18-40 actually represents.

-vests: please clarify why a single vest was not used for both EV and FV conditions. In considering the use of two different vests, how did the distribution of added load compare between the two vests? In particular I’m curious as to how high the load was on the torso under the two different vest conditions.

-static stance tests: Assuming the fixed bar was supported independently of the FootSTEPS device, did the authors confirm there was minimal difference in vertical loading with the use of the fixed bar to ensure participants were not leaning on the bar?

-Results

Figures: Are figures S1-S3 to be considered supplemental? Some pictures of the vests and testing scenario are key to understanding the methodology, however, S3 does not appear to add much beyond S1 and S3. If figure count limitations from the journal are an issue, it appears many of the currently separate figures in the manuscript could be combined to lessen the total figure count. For example, S1 and S2 could likely be combined into a ‘single figure’ with 4 images. Also of note, some of the figures’ images would benefit from additional cropping to limit the images to the points of interest (ex. there is a lot of ceiling unnecessarily displayed).

Another concern with the manuscript's figures is that Figures 1-5 do not appear to be adding information beyond what is displayed in the tables. Is there anything novel that is being displayed in those figures that is not evident in the manuscript’s tables?

-Discussion

-last sentence of section: ‘integrated’ should be ‘integrating’

6. PLOS authors have the option to publish the peer review history of their article (what does this mean?). If published, this will include your full peer review and any attached files.

Reviewer #1: **Yes: **Ram Kinker Mishra

Reviewer #2: No

---

## [Author Response · Author response to Decision Letter 0]

29 Dec 2020

Reviewer Responses:

We are very appreciative of the time the reviewers have taken to critique our manuscript. We have attempted to address all concerns and feel that the resulting changes have improved the paper. Below are point-by-point responses (red font) to each critique (regular type).

Reviewer #1:

1. Authors justified of excluding female based on distinct body fat accumulation tendency between male and female. However, in the current study objective of the authors is to compare effect of weight distribution conditions (i.e., Forward distributed vs Evenly distributed vs Normal weight) and not within subject differences. Therefore, in my opinion given justification to exclude female is weak and raises issues related to generalizability of the results.

• Author response: Our decision to exclude females was based on a desire to avoid any potential confounding influences due to sex. Although this was a within-subject study, there are noted differences in both body fat distributions and gait mechanics between sexes, and these could have introduced interaction effects. In addition, it was much easier to simulate male obesity than female obesity with our added body mass distribution methodology. We do think that there are enough potential sex differences to justify focusing only on one sex (alternately, we could have doubled our sample size and included sex as a variable). However, we completely agree that because of these potential differences, generalizability is limited and should have been explicitly stated as a limitation. To address this we have modified the following statement in the methods section: “In consideration of the distinct body fat accumulation tendencies [27] as well as movement differences [28] between males and female, we recruited only male participants.” We also added a limitation in the conclusions section: “Additionally, since we exclusively tested young male participants, our results may not fully generalized to females or elderly individuals, and factors related to these populations should also be considered.” 

2. Authors need to double check for the typo. For instance, in the abstract, there were typo such as (..).

• Author response: Apologies for the typos. English is not my native language. We have gone through the manuscript thoroughly.

3. Once the abbreviation of a term is provided than authors do not need to provide it again. For instance in the abstract authors provided abbreviation for evenly distributed (EV), or frontloaded vest (FV) twice.

• Author response: Apologies, this was an oversight on our part. Second definition has been deleted.

4. In the methodology, if the experiments were conducted according to declaration of Helsinki then authors should mention it in the methodology.

• Author response: According to your review, we added “This study was conducted in accordance with the declaration of Helsinki.”

5. In the figure, significant differences are not represented. The figures should be self-explanatory, and readers should not be needed to refer text or able to identify group significant differences. Providing significant differences through arrow would make it convenient for the readers to understand. Additionally, in figure 1, authors should have represented the AP_Static or ML_Static as AP Static or ML Static.

• Author response: Apologies – the significance markers somehow got left off during a format conversion. We have added these to each figure and modified the Figure 1 labels. 

6. In discussion, references are required for the statement, “A few previous studies have indirectly investigated the effects of mass on medial longitudinal arch deformation.”

• Author response: These references were provided in the specific sentences that followed this general sentence. However, since this wasn’t clear, we have added these references to the general sentence as well. 

7. The following sentence in the discussion was confusing and need to be rephrashed, “This reduced plantar cutaneous sensitivity has been associated with increased postural sway [7], thus our decision to simultaneously investigate postural sway measures.”

• Author response: The purpose of this sentence was to explain why we measured postural control. To be clear, we have modified to, “driving our decision to simultaneously investigate postural sway measures.”

8. It was not clear if the authors want to indicate association or causation in the following statement in discussion, “Previous load carriage studies show that healthy young individuals walking with a heavy backpack decrease stride length and increase double support time through decreased hip and knee sagittal plane angular displacement [44].”

• Author response: Thank you for this comment. In the sentence, we focused on the potential effects of body distribution on walking coordination during walking. To be clear, we have adjusted the word, “through,” to the word, “by,”: “Previous load carriage studies show that healthy young individuals walking with a heavy backpack decrease stride length and increase double support time by decreasing hip and knee sagittal plane angular displacement.” 

Reviewer #2:

Abstract - “However, the interactive effects between body mass and distribution may disrupt physical function and/or performance in other populations—such as elderly and individuals with obesity or diabetes.” I’m not sure this speculation is needed in the abstract.

• Author response: Thank you for this comment. We intended to provide some vision towards the applications that motivated our study. However, we agree that this borders on speculation, thus, we have modified the wording to: “However, as this is a first step study, follow-up studies are necessary to further support the clinical role of plantar shear in other populations such as elderly and individuals with obesity or diabetes.” 

Introduction

-“Changes in shear stresses are relatively higher compared to baseline values than changes in vertical pressure, and thus shear detecting mechanoreceptors may have a greater influence on relaying postural sway information needed for balance.” This sentence is not clear. Under what conditions or in what situations are changes in shear stresses expected to be relatively higher? Also are there any references that can be cited to support the premise the authors are trying to convey?

• Author response: We have modified this statement and added two references. It now reads: “Changes in shear stresses can be similar to changes in vertical pressure, thus having a relatively higher change compared to baseline values [24]. In addition, net shear forces are directly proportional to CoM-CoP differences and therefore carry phasic sway-relevant information needed for balance [24,25].” 

Methods

Participants: The first sentence reads as though a sample of participants with actual ages ranging from 18 to 40 years old were recruited. However, the 3rd sentence seems to suggest the previously named age range was an inclusion criterion as opposed to a description of the actual participants enrolled. Please clarify what 18-40 actually represents.

• Author response: Thank you for this comment. The inclusion criteria of the participants' age were from 18 to 40 years old because individuals in the age range have shown similar muscle strength and physical function. However our actual recruited participants were primarily from early to mid 20s. To avoid confusion, we have deleted the inclusion words, “between 18 to 40 years old,” in the sentence, and then we have adjusted the sentence as “A sample of 20 young healthy male participants were recruited...” 

Vests: please clarify why a single vest was not used for both EV and FV conditions. In considering the use of two different vests, how did the distribution of added load compare between the two vests? In particular I’m curious as to how high the load was on the torso under the two different vest conditions.

• Author response: In pilot testing, the vest for the EV condition was extremely uncomfortable around the neck when only loaded in the front. In addition, it did not easily accommodate sufficient weight. It has 40 pockets, each allowing one pound sandbag. So, the maximum load capacity of the vest is 40 lbs (20 lbs and 20 lbs to the front and the back, respectively). As the load of each vest condition was 20% of an individual's body weight, the vest for the EV condition could not be easily used for some participants whose 20% of body weight was over 20 lbs for the FV. Meanwhile, the vest for the FV condition was a baby carrier only allowing sandbags to be loaded in front. We matched the load distribution anteroposteriorly and superoinferiorly for the EV. Likewise, For the FV the load was evenly distributed in the front. We used a weighted box that can be placed in the area where a baby is in and a large pocket located at the anteroinferior aspect of the vest. In addition, we used small pockets containing a couple of sandbags at each side. The load distribution should be roughly visible from the pictures in figure 1, which was part of the reason for providing those. We have added a statement to the methods to help clarify the use of the baby carrier: “The baby carrier was used in place of the exercise vest because it better accommodated an asymmetrical load and was more comfortable around the shoulders and neck when fully front loaded.” 

Static stance tests: Assuming the fixed bar was supported independently of the FootSTEPS device, did the authors confirm there was minimal difference in vertical loading with the use of the fixed bar to ensure participants were not leaning on the bar?

• Author response: We did not formally analyze the vertical forces for inclusion in this manuscript, but pilot testing suggested that our instructions, practice trials, and visual checks were sufficient to minimize any leaning on the bar. We attached temporary markers vertically aligned to all participants once they stood uprightly on the FootSTEPS device for the Static stance test. Although the alignment was not perfectly vertical, all participants were instructed to keep aligning these markers in a line of grid that is shown on the screen showing their profile view during the test. While recording, one researcher kept checking any movements of these temporary markers, another researcher kept watching the posture of a participant on the side of the participant. If the aligned line was broken, we discarded that trial, we resumed the test. We’ve added a few words to the methods to better connect these instructions with the standardized posture: “To standardize posture as well as minimize reliance on the bar, the participant was instructed to watch a screen in front…”

Results

Figures: Are figures S1-S3 to be considered supplemental? Some pictures of the vests and testing scenario are key to understanding the methodology, however, S3 does not appear to add much beyond S1 and S3. If figure count limitations from the journal are an issue, it appears many of the currently separate figures in the manuscript could be combined to lessen the total figure count. For example, S1 and S2 could likely be combined into a ‘single figure’ with 4 images. Also of note, some of the figures’ images would benefit from additional cropping to limit the images to the points of interest (ex. there is a lot of ceiling unnecessarily displayed).

• Author response: Thank you for this comment. We originally included these as supplemental files. However, we agree that S3 does not add much, and so we have combined S1 with S2 and discarded S3. Then, we have put the combined one in the methods of the paper rather than as supplementary material and trimmed excess space. We appreciate this suggestion.

Another concern with the manuscript's figures is that Figures 1-5 do not appear to be adding information beyond what is displayed in the tables. Is there anything novel that is being displayed in those figures that is not evident in the manuscript’s tables?

• Author response: The graphs are based on the tables. As you mention, the information is overlapped. However, after some consideration, we still feel that the graphs provided visual information that makes it easier for readers to obtain and understand the results. For example, the graphs for AP CoP location independently provide the base line (the red line) that is not present in the tables. We’d like to include both in the paper.

Discussion

-last sentence of section: ‘integrated’ should be ‘integrating’

- Author response: Apologies for the typo. Corrected.

---

## [Decision Letter · Decision Letter 1]

22 Jan 2021

Added body mass alters plantar shear stresses, postural control, and gait Kinetics: Implications for obesity.

PONE-D-20-30951R1

Dear Dr. Jeong,

We’re pleased to inform you that your manuscript has been judged scientifically suitable for publication and will be formally accepted for publication once it meets all outstanding technical requirements.

Kind regards,

Bijan Najafi

Academic Editor

PLOS ONE

Additional Editor Comments (optional):

Reviewers' comments:

Reviewer's Responses to Questions

**Comments to the Author**

1. If the authors have adequately addressed your comments raised in a previous round of review and you feel that this manuscript is now acceptable for publication, you may indicate that here to bypass the “Comments to the Author” section, enter your conflict of interest statement in the “Confidential to Editor” section, and submit your "Accept" recommendation.

Reviewer #2: All comments have been addressed

2. Is the manuscript technically sound, and do the data support the conclusions?

Reviewer #2: (No Response)

3. Has the statistical analysis been performed appropriately and rigorously? 

Reviewer #2: (No Response)

4. Have the authors made all data underlying the findings in their manuscript fully available?

Reviewer #2: (No Response)

5. Is the manuscript presented in an intelligible fashion and written in standard English?

Reviewer #2: (No Response)

6. Review Comments to the Author

Reviewer #2: (No Response)

7. PLOS authors have the option to publish the peer review history of their article (what does this mean?). If published, this will include your full peer review and any attached files.

Reviewer #2: No

---

## [Editor Report · Acceptance letter]

26 Jan 2021

PONE-D-20-30951R1 

Added body mass alters plantar shear stresses, postural control, and gait Kinetics: Implications for obesity. 

Dear Dr. Jeong:

I'm pleased to inform you that your manuscript has been deemed suitable for publication in PLOS ONE. Congratulations! Your manuscript is now with our production department. 

Kind regards, 

on behalf of

Dr. Bijan Najafi 

Academic Editor

PLOS ONE